# Comparing the Phylogenetic Distribution of Multilocus Sequence Typing, Staphylococcal Protein A, and Staphylococcal Cassette Chromosome Mec Types in Methicillin-Resistant Staphylococcus Aureus (MRSA) in Korea from 1994 to 2020

**DOI:** 10.3390/antibiotics12091397

**Published:** 2023-09-01

**Authors:** You-Jin Hwang

**Affiliations:** 1Department of Biomedical Engineering, Gachon University, Incheon 21936, Republic of Korea; gene@gachon.ac.kr or ghilhwang@naver.com; Tel.: +82-032-820-4545; Fax: +82-032-820-4449; 2Department of Health Sciences and Technology (GAIHST), Gachon University, Incheon 21999, Republic of Korea

**Keywords:** MRSA, MLST, phylogenetic distribution, *SCCmec* types, staphylococcal protein A (*spa*) typing

## Abstract

*Staphylococcus aureus* (*S. aureus*) bacteremia is one of the most frequent and severe bacterial infections worldwide. Methicillin-resistant *Staphylococcus aureus* (MRSA) is a serious human pathogen that can cause a wide variety of infections. Comparative genetic analyses have shown that despite the existence of a vast number of genotypes, genotypes are restricted to certain geographical locations. By comparing multilocus sequence typing (MLST) and *SCCmec* types from 1994 to 2020, the present study intended to discover which genotype genes were related to MRSA infections. MLST, *Staphylococcus aureus* protein A (spa), and *SCCmec* typings were performed to determine their relationship during those years. Results revealed that MRSA isolates in the Republic of Korea were distributed among all major staphylococcal cassette chromosome *mec* (*SCCmec*) types. The majority of *SCCmec* isolates belonged to *SCCmec* type II and type IV. The majority of MLST had the sequence type (ST) 72, 239, 8, or 188. By contrast, minorities belonged to ST22 (SCC*mec* IV), ST772 (SCC*mec* V), and ST672 (SCC*mec* V) genotypes. The *SCCmec* type was determined for various types. The *spa* type was dispersed, seemingly regardless of its multidrug resistance property. The MLST type was found to be similar to the existing typical type. These results showed some correlations between resistance characteristics and types according to the characteristics of the MLST types distributed, compared to previous papers. Reports on genotype distribution of MLST and *SCCmec* types in MRSA are rare. These results show a clear distribution of MLST and *SCCmec* types of MRSA from 1994 to 2020 in the Republic of Korea.

## 1. Introduction

*Staphylococcus aureus* bacteremia is one of the most frequent and severe bacterial infections worldwide. Methicillin-resistant *Staphylococcus aureus* (MRSA) is a serious human pathogen that can cause a wide variety of infections [1]. MRSA was first described in the UK in 1961. Thereafter, it was reported in Europe in 1965, Australia in 1966, the USA in 1968, and Asia in the 1970s [2,3]. Clinical research on MRSA was conducted in the early 1990s in the Republic of Korea. Recently, the area of MRSA research is growing [4,5,6,7]. Due to its unique properties, MRSA has become one of the most significant nosocomial pathogens worldwide today. MRSA can be genotyped by one or more of several molecular techniques, including pulsed-field gel electrophoresis (PFGE) [8], multilocus sequence typing (MLST) [9] (https://pubmlst.org/organisms/staphylococcus-aureusn, accessed on 1 May 2020), and staphylococcal protein A (*spa*) typing [10] (http://spaserver.ridom.de, accessed on 1 May 2020).

While SCC*mec* typing has become essential for the characterization of MRSA clones in epidemiological studies, there are two issues, (1) assigning SCC*mec* elements, and (2) naming novel elements or variants. The methicillin resistant structural staphylococcal cassette chromosome *mec* (SCC*mec*) gene (SCC *mecA)* is a small (2007bp) part of a much larger genetic element that is inserted precisely into the *S. aureus* chromosome [11]. MRSA can also be classified through SCC*mec* typing, which involves PCR sequencing test for one or more of 11 SCC*mec* types (I through XI, some further subclassified into A, B, C1, C2, and E) (https://www.sccmec.org/index.php/en, accessed on 1 May 2020) [12]. Different typing techniques can be employed for different purposes. MLST can be applied to phylogenetic analyses. It can also be used for longitudinal and long-term studies [13]. MLST involves PCR amplification and sequencing of seven housekeeping genes followed by allelic profiling and assigning sequence types (STs) to each strain.

Sequences of our results are compared to known alleles at each locus via the MLST website (https://pubmlst.org/organisms/staphylococcus-aureus, accessed on 1 May 2020), where every isolate is described by a seven-integer allelic profile that defines a sequence type (ST). The objective of this study was to compare MLST and SCC*mec* typing to show the complexity of evolutionary events using isolates collected from 1994 to 2020 in the Republic of Korea.

## 2. Materials and Methods

### 2.1. Materials and Bacterial Isolates

A total of 134 *S. aureus* strains were obtained from clinical patients at Gachon University Gil Medical Center in Incheon, Republic of Korea, between April 2016 and June 2019 [14,15]. It was approved by the ethics committee of Gil Hospital, Gachon University of Medicine. Sample identification and antimicrobial susceptibility testing of *S. aureus* isolated from blood culture were performed using a MicroScan Pos Breakpoint Combo panel type 28 (PBC28; Beckman Coulter, West Sacramento, CA, USA). Sample strains were streaked onto sheep blood agar (Sinyang Diagnostics, Seoul, Republic of Korea) and transported to our laboratory after cultivation.

### 2.2. Antimicrobial Susceptibility Testing

Antimicrobial susceptibility testing was performed using the Kirby–Bauer disc diffusion method described by the Clinical and Laboratory Standard Institute (CLSI) guidelines, 2013 [16]. Each bacterial suspension was adjusted to a McFarland 0.5 turbidity, swabbed onto Muller–Hinton agar, and incubated in the presence of antibiotic discs at 35 °C for 18 h. This test used the following 19 antibiotic discs (Liofilchem, Roseto degli Aburzzi, Italy): penicillin G (10 IU), methicillin (5 μg), kanamycin (30 μg), gentamicin (10 μg), streptomycin (10 μg), tetracycline (30 μg), erythromycin (15 μg), vancomycin (30 μg), chloramphenicol (30 μg), amoxicillin (25 μg), ticarcillin (75 μg), piperacillin (100 μg), cefepime (30 μg), cefotaxime (30 μg), ceftazidime (30 μg), imipenem (10 μg), ertapenem (10 μg), meropenem (10 μg), and aztreonam (30 μg). Diameters of the inhibition zones were measured and each isolate was determined as resistant or susceptible to antimicrobial agents based on the CLSI guidelines and Liofilchem quality control parameters. *S. aureus* control strain *Staphylococcus aureus* ATCC 29213 was obtained from the Korean Culture Center of Microorganisms, Seodaemun-gu, Seoul, Republic of Korea. Antimicrobial susceptibility test was performed according to an existing procedure [14,15].

### 2.3. Identifying mecA, bla_TEM_, and SCCmec Typing by Multiplex Real-Time PCR

PCR primers used to detect *mec*A and *bla*_TEM_ genes are listed in Table 1 [14,15,17,18,19]. The following reaction mixture was added to each sample: 10 pmol of each primer, 2 μL DNA (100 ng), and 10 μL iQ^TM^ SYBR^®^ Green supermix (2× reaction buffer with dNTPs, iTaq DNA polymerase, SYBR^®^ Green I, fluorescein, and stabilizers, Bio-Rad, Hercules, CA, USA). The volume was adjusted to 20 μL by adding autoclaved triple-distilled water. PCR cycling conditions on a thermal cycler (iQ5, Bio-Rad and TC-512, Hercules, Californaia, USA) were as follows: 94 °C for 3 min followed by 35 cycles of denaturation at 94 °C for 30 s, annealing at 56 °C for 30 s, and extension at 72 °C for 45 s. The reaction was ended with a final extension step at 72 °C for 10 min. Multiplex PCR was carried out for SCC*mec* typing using nine pairs of primers specific for SCC*mec* types I, II, III, IVa, IVA, IVb, IVc, IVd, and V [14,15,17,18,19]. PCR products were subjected to electrophoresis using 2% agarose gel in 1× Tris-borate-EDTA (TBE) buffer at 100 V for 25 min. A 100 bp DNA ladder (Bioneer, Daejeon, Republic of Korea) was used as a molecular size maker. PCR products in gels were then visualized with a Safe Green loading dye (Applied Biological Materials Inc, Vancouver, Canada).

### 2.4. Determination of spa and Multi-Locus Sequence Typing (MLST) Sequence Types

Spa typing was performed as described by Harmsen et al. [10,20]. The polymorphic X region of the *spa* gene was amplified using primers spa1095F (5′-AGACGATCCTTCGGTGAGC-3′) and spa1517R (5′-GCTTTTGCAATGTCATTTACTG-3′). PCR spa gene products were subjected to DNA sequencing of both strands by Bioneer (Bioneer, Daejeon, Republic of Korea). Sequences were analyzed using Ridom StaphType v2.0.3 software (Ridom GmbH). Guidelines derived from the Ridom SpaServer database (http://www.spaserver.ridom.de, accessed on 1 May 2020) were used to assign edited sequences to particular spa types. Relationships between spa types were investigated using the based-upon repeat pattern (BURP) clustering algorithm [21] and incorporated into Ridom StaphType. Sequences were analyzed using the multiple sequence alignment of the CLUSTALW program (https://www.genome.jp/tools-bin/clustalw, accessed on 1 May 2020, Kyoto University Bioinformatics Center).

Our MLST was performed as previously described [14,15], using the *S. aureus* MLST database at http://www.mlst.net/, accessed on 1 May 2020). Primers designed for the seven multi-locus sequence typing (MLST) housekeeping genes (*arcC*, *aroE*, *glpF*, *gmk*, *pta*, *tpi*, and *yqiL*) were obtained from the MLST database (http://www.mlst.net/, accessed on 1 May 2020). PCR testing was performed using 10 pmol of upstream primer, 10 pmol of downstream primer, 100 ng/mL of template, and 10 uL of 2× iQ^TM^ SYBR^®^ Green supermix (Bio-Rad, Hercules, CA, USA). Sterile water was added to achieve a volume of 20 uL. PCR cycling conditions were as follows: 95 °C for 5 min; followed by 30 cycles of 94 °C for 30 s; 55 °C for 30 s; 72 °C for 1 min; and a final extension step of 72 °C for 10 min. PCR products of seven housekeeping gene fragments were sequenced (Bioneer, Daejeon, Republic of Korea) and compared with allele profiles of *S. aureus* samples [14,15]. MLST database (http://www.mlst.net/, accessed on 1 May 2020) and sequence types (STs) were derived and analyzed with eBURST software (http://saureus.mlst.net/eburst/, accessed on 1 May 2020). Sequences of unknown alleles were confirmed by repeating the analysis procedure submitted to the MLST database [14,15,20].

### 2.5. Comparative Analysis of MLST and SCCmec Types and the Epidemiological Survey of 1994–2020

These analyses compared allelic divergence in a predominantly diversified population, which would result in 134 MLST types within the previous results from 1994 to 2020 in the Republic of Korea, and then compared these with 22 SCC*mec* results by result type correlation [14,15,22,23,24]. Two data sets based on either MLST housekeeping genes and/or SCC*mec*A genes were combined.

## 3. Results

### 3.1. Results of Antimicrobial Susceptibility Testing and Multi-Drug Resistance Genes

The results of the antibiotic resistance testing for nine bacteria strains are shown in Table 1 and Table 2. It was found after testing 19 types of antibiotics that gh 13 and gh 90 bacteria showed resistance to 12 and 15 antibiotics, respectively. The results of the test of carbapenem antibiotic resistance also showed this for strains gh65, gh68, gh84, and gh90 out of gh13. The gh13 strain showed resistance to methicillin, penicillin, and erythromycin. It was resistant to nine carbapenems (ticarcillin (75 μg), piperacillin (100 μg), cefepime (30 μg), cefotaxime (30 μg), ceftazidime (30 μg), imipenem (10 μg), ertapenem (10 μg), meropenem (10 μg), and aztreonam (30 μg)) for *S. aureus* gh13, gh65, and gh90 species. It was found that gh90 and gh13 strains had similar resistance. They were both resistant to kanamycin, gentamicin, and streptomycin. Gh68 and gh84 also showed similar resistance to antibiotics. Gh54 was resistant to only one antibiotic. Bacteria separated from different patients even showed similar antibiotic resistance patterns. It was confirmed that nine bacteria strains showed the same antibiotic resistance characteristics (Table 1 and Table 2).

### 3.2. Occurrence of S. aureus MLST types in the Republic of Korea

These results were observable in the pattern of resistant genes, SCC*mec* types, and MLST types analyses for the antibiotic resistance of nine strains, and in the samples previously analyzed with 13 strains [14] (Table 3).

Of nine strains, gh6, gh68, and gh84 showed the same MLST type. Results for the gh38 strain were similar to previous strain results [14,15]. However, *arc* and *glpF* were different in MLST. For gh90, *grnK*, *pta*, *tpi*, and *yqiL* were different from previous MLST results. However, gh13 had no similarities with other strain results. Gh49 and gh54 showed different aspects of *aroE.* Gh65 showed similarities in *glpF*. However, the rest were new MLST. As a result, the MLST types for nine strains appeared to be new. They have not yet been reported (Table 3).

### 3.3. Screening Results of MLST, spa, and SCCmecA Types Analysis and Phylogenetic Tree

KoreaMed, Medline, PubMed, MLST database, and Google Scholar were searched using the terms ‘MRSA’, ‘genotype’, ‘MLST’, ‘spa’, and ‘SCC*mec* types’ and/or ‘Republic of Korea or/and Korean’. A total of 73 abstracts and 45 manuscripts published between 1994 and 2020 were identified for inclusion in this study. After reviewing all relevant abstracts and articles, 86 full-length articles containing genotyping information based on at least one of the methodologies described above were considered for results [24,25,26]. Comparisons of genetic distance between strains were expressed as genotype of spa and MLST analysis, as shown in Figure 1.

These results showed concentration in a specific area. They showed a few changes in MLST type IV. MLST type II was concentrated around the center. However, some were distributed in different locations (Figure 1).

Phylogenetic tree results were obtained by analyzing some of the papers reported results from the Republic of Korea. As a result, MLST types II and IV showed a wide range distribution overall. In particular, MLST type II was widely distributed overall, whereas MLST type IV showed a narrow and central range of locations (Figure 2). These results showed regional differences, infection routes, genetic connections, and other various causes. The results from 22 strains showed MLST types I, II, III, and IV. The remaining other types were not shown. Strains gh2, gh13, gh19, gh21, and gh90 had confirmed resistance to nine carbapenem antibiotics. Most of these strains were found to be MLST type II (Figure 1).

## 4. Discussion

The number of reports on MRSA has gradually increased in the Republic of Korea over the last twenty years [26]. In the Republic of Korea, types ST5 and ST239 were found to be the two predominant MRSA clones in Korean hospitals from 1994 to 2006 [24,26,27,28,29]. The majority of ST5 strains carried SCC*mec* II elements, whereas most of the ST239 strains carried SCC*mec* III or SCC*mec* IIIA [24,26,27].

MRSA clones were first described in hospital patients and nosocomial in aetiology. It has been reported that global HA-MRSA clones including CC5-SCCmecII (USA100), CC5-SCCmecIV (USA800), CC8-SCCmecIV (USA500), CC22-SCCmecIV (EMRSA-15), CC30-SCCmecII (EMRSA-16), CC45- SCC*mec* IV, and ST239-SCCmecIII [30] harbor SCC*mec* elements of type I, II, or III [31]. Descriptions of STs that share at least five out of seven alleles are further grouped into clonal complexes (CC) [32].

From 1996 to 2000, there were many MLST reports of MRSA clones, mainly on types II and IV [29,33]. Other types such as III, IV, VII, and VIII began to be reported gradually [26]. From 2001 to 2006, there were many reports of types III and IIIA. After that, reports decreased. Around 2005, there were many reports of types III, IIIA, and IV [34,35,36]. From 2006 to 2010, reports of type IV and various MLST types generally occurred [36,37,38].

These results were found to show differences depending on the sample area and year of sample collection. Reports of type I began to appear between 2007 and 2011 [6,39]. Since 2010, data that detailed analysis on types such as I, II, III, IV, Va, have been mainly classified. Type VII and VIII were reported twice in 1999 and 2014 in the Republic of Korea [7,40].

Another clone present in our study, ST5/SCC*mec* II, was previously characterized as a USA100 clone (New York/Japan Clone) [41]. ST5-SCC*mec* II was also reported in European countries such as Hungary, Portugal, and Austria [42]. Strain maps of isolates from Asia and the Pacific are especially diverse, with ST72 (CC8) being well described in the Republic of Korea, ST8 or ST30 in Japan, and ST59 in Taiwan, while an even greater diversity is present in China [43,44,45]. The Regional Resistance Surveillance Program monitored susceptibility rates and resistance development by geographic region, including 12 Asia-Pacific (APAC) countries. Among *Staphylococcus aureus* isolates, 37% were methicillin-resistant *S. aureus* (MRSA), with the highest occurrence in the Republic of Korea (73%) [46,47,48].

Since 2016, types I, II, III, and IV have mostly appeared [49]. From 2017 to 2019, types II, IV, and V mostly appeared. Recently, in 2020, there were several reports focusing on types II, IV, and V [50,51]. Moreover, the more variable markers frequently do not reflect the pathogen’s evolutionary history. Hence, they might provide potentially misleading information about the pathogen’s spread. More recent studies have demonstrated that staphylococcal evolution proceeds sufficiently fast that the dynamics of *S. aureus’s* spatial spread can be elucidated in great detail on the basis of genome-wide single-nucleotide polymorphisms [52,53].

Recent research results are trying to analyze resistant genes and their characteristics by combining more diverse analysis methods for these various analysis results [53,54,55,56]. It cannot be concluded that the data include all results reported in Korea. There might be some omissions in this result. However, we tried to collect and organize as much data surveyed in Korea as possible. Data papers on resistance specificity are being published [6,14,15,32,50,51]. Until recently, characteristics of SCCmec types have been reported in Korea from type I to type VIII [14,32,40,53,54,55,57]. However, there are no reports about the detection of pressure from other types such as type VI yet. Strains reported in Korea have mainly been reported following the recent emergence of new mutations in existing types. In addition, types II, III, and IV are detected among existing types. 

This study and results, based on existing reports from the Republic of Korea, reveal that the SCCmec type mainly shows multi-drug resistance. While the degree of resistance was weak for type I, the distribution of type III showed a degree of focus.

Similar studies should be carried out continuously.

## 5. Conclusions

This study obtained results by comparing 22 self-reported results with unknown MLST types in the Republic of Korea, which have been published annually since 1996, and analyzing the association between the SCC*mec* types [14,15,40].

In early 1996, when data were first reported in a study, a lot of MLST ST5, ST239, ST8, and ST72 were detected. In particular, type ST5 has several SCC*mec*II types, and type ST72 has several SCC*mec*IV types [9,38]. Recent reports have shown that ST5, ST72, ST188, and ST8 account for 45% of total cases, with various other types in addition to the existing results.

Similar to the results of this study, t2460 and t008 have been often detected recently. Types II and IV account for 60% of SCC*mec.* Types I and III partially appeared. The remaining types have been detected in small numbers.

## Figures and Tables

**Figure 1 antibiotics-12-01397-f001:**
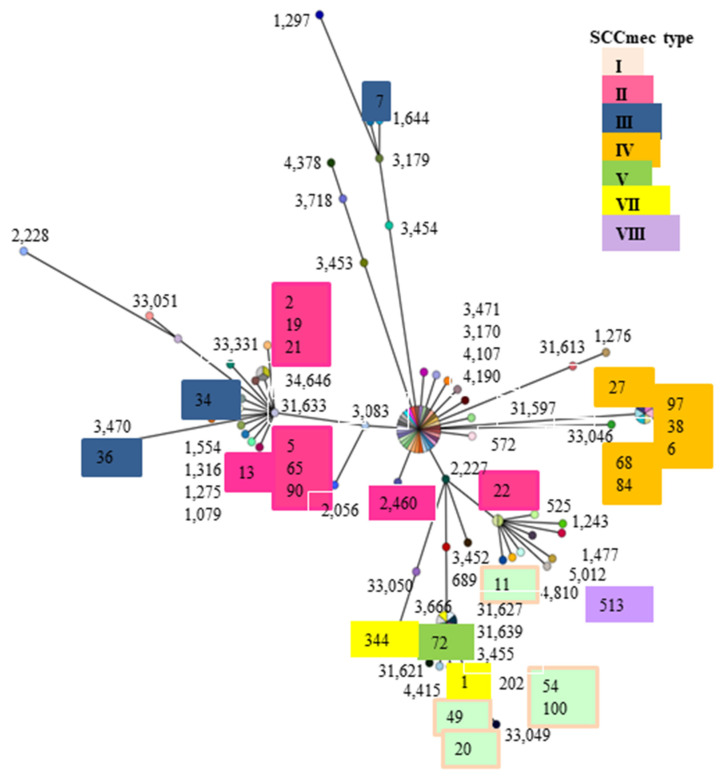
The distance comparisons were repeated by independently assessed spa types and *SCCmec* types samples of phylogenetic trees. OR distances were pooled for MLST types for the analysis. No is spa types ID number, color marks are *SCCmec* types, and diagrams are MLST distance. Color boxes are our results. Arabian numbers are quarry sequences with MLST samples.

**Figure 2 antibiotics-12-01397-f002:**
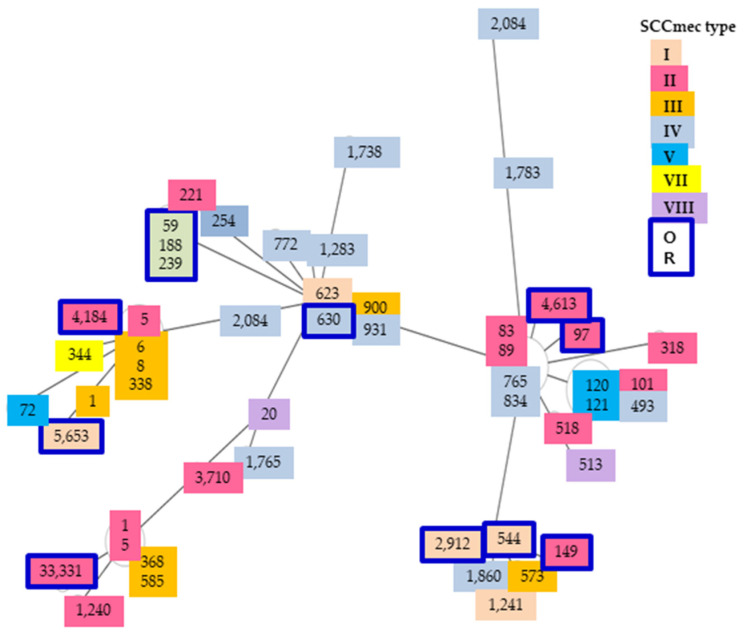
The distance comparisons were phylogenetic trees, organized by independent spa type and SCCmec type samples from 1996 to 2020 in the Republic of Korea. OR distances were pooled as MLST types for the analysis. No is spa types ID number, color marks are SCCmec types, and diagrams are MLST distance. Color boxes are our results. Arabian numbers are quarry sequences with MLST samples.

**Table 1 antibiotics-12-01397-t001:** Phenotypic antibiotic resistance patterns in *S. aureus* samples.

*S. aureus*/Samples	*Meth	Pen	Kan	Erh	Gen	Tet	Strep	Van	Chlo	AML	TC	PRL	FEP	CTX	CAZ	IMI	ETP	MRP	ATM
Gh6 ^†^	R	R	R	S	S	S	I	S	S	S	S	I	S	S	S	S	S	S	R
Gh13	R	R	I	R	S	I	I	S	S	I	R	R	R	R	R	R	R	R	R
Gh38	R	R	R	S	R	S	I	S	S	S	I	I	S	S	S	S	S	S	R
Gh49	S	S	I	S	S	S	I	S	S	S	S	I	S	S	S	S	S	S	R
Gh54	S	R	S	S	S	S	S	S	S	S	S	S	S	S	S	S	S	S	R
Gh65	R	S	S	R	S	I	I	S	S	S	I	I	I	I	R	R	R	R	R
Gh68	R	S	R	S	R	S	R	S	S	I	I	I	I	R	I	S	S	I	R
Gh84	R	S	R	S	S	S	R	S	S	I	I	I	I	I	R	S	I	I	R
Gh90	R	R	R	R	R	I	R	S	S	S	R	R	R	R	R	R	R	R	R

We tested the following 19 antibiotic discs (Liofilchem, Roseto degli Aburzzi, Italy). We measured the diameter of the inhibition zones and determined each isolate as resistant or susceptible to antimicrobial agents based on CLSI guidelines and criteria, and Liofilchem quality control. ^†^ gh is 9/134 samples in Gil hospital *Meth is methicillin, Pen is penicillin, Kan is kanamycin, Gen is gentamicin, Tet is tetracycline, Strep is streptomycin, Van is vancomycin, Chlo is chloramphenicol, AML is amoxicillin, TC is Ticarcillin, PRL is piperacillin, FEP is cefepime, CTX is cefotaxime, CAZ is ceftazidime, IMI is imipenem, ETP is ertapenem, MRP is meropenem, and ATM is aztreonam.

**Table 2 antibiotics-12-01397-t002:** Genotypic antibiotic resistance gene patterns in *S. aureus.*

*S. aureus*/Strains	MRSA/MSSA	*TEM*	*mecA*	*SCCmec* TypeII/VI	*aac6-aph2*	*tetM*	*ermA/C*
Gh6 *	MRSA	pos	pos	IV	ND	ND	ND
Gh13	MRSA	pos	pos	II	ND	pos	pos
Gh38	MRSA	pos	pos	IV	pos	ND	ND
Gh49	ND	ND	ND	nd(I)	ND	ND	ND
Gh54	ND	pos	ND	nd(I)	ND	ND	ND
Gh65	MRSA	pos	pos	II	ND	pos	pos
Gh68	MRSA	pos	pos	nd(IV)	pos	ND	ND
Gh84	MRSA	pos	pos	nd(IV)	ND	ND	ND
Gh90	MRSA	pos	pos	II	pos	pos	pos

* gh is 9/134 new types sampled in Gil hospital, pos is positive, ND is not determined, nd is not completed gene typing, *TEM* is *blaTEM* gene, *mecA* is methicillin gene, *aac6-aph2* is gentamicin gene, and *ermA*/C is erythromycin gene.

**Table 3 antibiotics-12-01397-t003:** Comparison of Novel MLST types in *S. aureus* samples.

Isolate	*arcC*	*aroE*	*glpF*	*gmk*	*pta*	*tpi*	*yqiL*	Collection ^1^	*SCCmec*	Reference
gh6	1	601	549	8	4	4	3	V	IV	new
gh13	1	511	549	72	1	56	10	Wb	II	new
gh38	3	601	149	8	4	4	3	UK	IV	new
gh49	3	404	549	8	4	1	1	cms	I	new
gh54	3	432	549	8	4	1	1	U	I	new
gh65	1	864	549	72	12	1	10	Sp	II	new
gh68	1	601	549	8	4	4	3	Sp	IV	new
gh84	1	601	549	8	4	4	3	Wb	IV	new
gh90	1	601	549	72	12	1	10	Wb	II	new
gh11	1	4	1	185	4	497	76	Sp	I	[14] ^2^
gh2	1	4	1	4	559	495	10	Sp	II	[14]
gh5	1	4	1	4	559	41	10	Sp	II	[14]
gh7	2	2	95	185	6	201	500	Vag	III	[14]
gh19	1	4	1	4	559	134	10	Sp	II	[14]
gh20	3	1	1	8	322	495	295	Wb	I	[14]
gh21	1	4	1	4	559	495	10	Sp	II	[14]
gh22	1	4	1	4	559	495	10	Sp	II	[14]
gh27	3	696	795	4	4	394	3	UK	IV	[14]
gh34	1	4	1	8	4	497	76	UK	III	[14]
gh36	177	4	1	8	4	368	76	UK	III	[14]
gh97	1	4	1	8	4	497	3	Wb	IV	[14]
gh100	3	1	1	8	1	134	295	Wb	I	[14]

Comparison of 22 MLST type samples. ^1^. Samples collection in Wb (web), SP (sputum), v (vesicle swab), uk (urethra swab), cms (maxillary sinus swab), u (urine), vag (vaginal swab). ^2^. No 14 are the samples that have previously been analyzed with 13 strains [14] (Table 3).

## Data Availability

Data are contained within the article.

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
