# Peer review of "Comparing the Phylogenetic Distribution of Multilocus Sequence Typing, Staphylococcal Protein A, and Staphylococcal Cassette Chromosome Mec Types in Methicillin-Resistant Staphylococcus Aureus (MRSA) in Korea from 1994 to 2020"

_antibiotics, 2023, doi:10.3390/antibiotics12091397_

Round 1

Reviewer 1 Report

Comments:

The manuscript prepared by You Jin Hwang describes the genotypic characteristics of MRSA strains isolated from blood infections in Korea. It provides important data regarding the distribution of MRSA SCCmec types during 2016 - 2019, collected from Gil Hospital (Korea). Overall, however, the manuscript needs revisions. Specific comments:

Please verify the title.

Materials and methods section:

à  Please add the characteristics of the hospital (type, number of beds), the number of S. aureus/MRSA isolates collected during 2016-2019. The number of isolates is mentioned in the Result section (lines 133-134), however it should be clearer presented.

à  The reference standard used for antibiotic susceptibility testing, the Clinical and Laboratory Standard Institute (CLSI) guidelines, 2013 is an old version and the author should refer to a more recent version.

Results section needs major revision

à  Lines 135-140: remove the concentration of antibiotics. Please check the standard and review the carbapenem antibiotics. In the current study the researcher tested three carbapenems (IMP, MEM, ETP) Please add the species name to the ID of the strain (i.e. S. aureus gh..)

à  Table 1: the results (interpretation, i.e categories S, R or I) of the antibiotic susceptibility testing (AST) should be included, not the diameters of inhibition zones. The main results of the AST (phenotypically assessment) are not presented in the text. The author should mention the MRSA or MSSA types, VRSA/VSSA identified. Vancomycin susceptibility of the strains should be evaluated using E-test or microdilution test, according to the CLSI standard.

à  Lines 158-160: the sentence is not clear, please revise.

à  Tables 1 and 2: please add what strains are, not just the code/ID.

à  Table 3 is not clear; the gp is missing from the name of the strain is not clear or complete. It is not clear for me which are the strains from the current study. Also, there are letters in the collection column that are not explained.

à  Please verify line 190.

Discussion section:

à  Lines 247,

à  Line 255, please explain: degree of resistance was weak

Conclusion section:

à  Lines 258-260, the information is not clear for me. I do not understand 22-self results?

à  Lines 261-264, what does a lot mean?, I do not understand a study? Whom study?

à  Please add the significance (importance, impact) of the study, the overall importance of these results, in general.

Thank you very much

Author Response

Response to the Comments 1 & 2

[Antibiotics] No. 2569477

Article Title: “Compare Phylogenetic distribution for MLST, spa, and SCCmec types in methicillin-resistant Staphylococcus aureus (MRSA) from 1994 to 2020 years in Korea”

Open Review

Quality of English Language

(x) I am not qualified to assess the quality of English in this paper
( ) English very difficult to understand/incomprehensible
( ) Extensive editing of English language required
( ) Moderate editing of English language required
( ) Minor editing of English language required
( ) English language fine. No issues detected

Yes

Can be improved

Must be improved

Not applicable

Does the introduction provide sufficient background and include all relevant references?

(x)

( )

( )

( )

Are all the cited references relevant to the research?

(x)

( )

( )

( )

Is the research design appropriate?

(x)

( )

( )

( )

Are the methods adequately described?

(x)

( )

( )

( )

Are the results clearly presented?

( )

( )

(x)

( )

Are the conclusions supported by the results?

( )

( )

(x)

( )

Comments and Suggestions for Authors

Comments:

The manuscript prepared by You Jin Hwang describes the genotypic characteristics of MRSA strains isolated from blood infections in Korea. It provides important data regarding the distribution of MRSA SCCmec types during 2016 - 2019, collected from Gil Hospital (Korea). Overall, however, the manuscript needs revisions. Specific comments:

Please verify the title.

Reviewers comments:

Materials and methods section:

à  Please add the characteristics of the hospital (type, number of beds), the number of S. aureus/MRSA isolates collected during 2016-2019. The number of isolates is mentioned in the Result section (lines 133-134), however it should be clearer presented.

Response #1. Thank for your careful consideration of the manuscript.

I have added final MRSA isolates number.

A total of 134 S. aureus strains were obtained from clinical patients at Gachon University Gil Medical Center

(reference 14, 15)

à  The reference standard used for antibiotic susceptibility testing, the Clinical and Laboratory Standard Institute (CLSI) guidelines, 2013 is an old version and the author should refer to a more recent version.

Results section needs major revision

Response #2. Thank for your careful consideration of the manuscript.

I have changed reference add recommended by reviewer no 16.

  1. Wayne, P.A.; CLSI. 2018. Methods for dilution antimicrobial susceptibility tests for bacteria that grow aerobically, 11th ed. Clinical and Laboratory Standards Institute, 11th ed.; 2018.

à  Lines 135-140: remove the concentration of antibiotics. Please check the standard and review the carbapenem antibiotics. In the current study the researcher tested three carbapenems (IMP, MEM, ETP) Please add the species name to the ID of the strain (i.e. S. aureus gh..)

Response #3. Thank for your careful consideration of the manuscript.

I have added recommended by reviewer.

for S. aureus gh 13, 65, 90 species.

à  Table 1: the results (interpretation, i.e categories S, R or I) of the antibiotic susceptibility testing (AST) should be included, not the diameters of inhibition zones. The main results of the AST (phenotypically assessment) are not presented in the text. The author should mention the MRSA or MSSA types, VRSA/VSSA identified. Vancomycin susceptibility of the strains should be evaluated using E-test or microdilution test, according to the CLSI standard.

Response #4. Thank for your careful consideration of the manuscript.

The results of the first antibiotic sensitivity test were hospital and obtained from the results and used.

Line 64-67. It was approved by the ethics committee of Gil Hospital, Gachon University of Medicine. Sample identification and antimicrobial susceptibility testing of S. aureus isolated from blood culture were performed using a MicroScan Pos Breakpoint Combo panel type 28 (PBC28; Beckman Coulter, West Sacramento, CA, USA).

à  Lines 158-160: the sentence is not clear, please revise.

Response #5. Thank for your careful consideration of the manuscript.

I have changed part. add recommended by reviewer no 158-160.

These results were showed in the pattern of resistant genes, SCCmec types, and MLST types analyzing for the antibiotic resistance of nine and the samples have previously analyzed with 13 strains reference 14 (Table 3).

à  Tables 1 and 2: please add what strains are, not just the code/ID.

Response #6. Thank for your careful consideration of the manuscript.

I have changed and add recommended by reviewer table 1 and 2.

  1. aureus /Strains ID no etc

Table 1. Phenotypic antibiotic resistance patterns in S. aureus samples.

Table 2. Genotypic antibiotic resistance gene patterns in S. aureus.

à  Table 3 is not clear; the gp is missing from the name of the strain is not clear or complete. It is not clear for me which are the strains from the current study. Also, there are letters in the collection column that are not explained.

Response #7. Thank for your careful consideration of the manuscript.

I have changed and add recommended by reviewer table 3.

Missing name anded gh 6 etc.

  1. Samples collection in Wb(web), SP(sputum), v(vesicle swab), uk(urethra swab), cms(maxillary sinus swab), u(urine), vag(vaginal swab). 2. No 14 is the samples have previously analyzed with 13 strains reference 14 Mun & Hwang, 2019, Antibiotics (Table 3).

à  Please verify line 190.

Discussion section:

Response #8. Thank for your careful consideration of the manuscript.

I have changed and add recommended by reviewer.

This study should be carried out continuously in the future.

à  Lines 247,

à  Line 255, please explain: degree of resistance was weak

Conclusion section:

Response #9. Thank for your careful consideration of the manuscript.

I have changed part.

Data papers on resistance specificity are being published [6, 14, 15, 32, 40, 48, 50, 51, 54]. Until recently, characteristics of SCCmec types have been reported in Korea from type I to type VIII [14, 32, 40, 50]. However, there are no reports about the detection of other pressure VI yet.

à  Lines 258-260, the information is not clear for me. I do not understand 22-self results?

Response #10. Thank for your careful consideration of the manuscript.

I have changed part.

This part explains was my individual comment.

I have this part took out.

à  Lines 261-264, what does a lot mean?, I do not understand a study? Whom study?

Response #11. Thank for your careful consideration of the manuscript.

I have changed the part.

This part explains was my individual comment.

I have this part took out.

à  Please add the significance (importance, impact) of the study, the overall importance of these results, in general.

Response #12. Thank for your careful consideration of the manuscript.

I have changed the part.

However, we tried to collect and organize as much data surveyed in Korea as possible. Data papers on resistance specificity are being published [6, 14, 15, 32, 40, 48, 50, 51, 54]. Until recently, characteristics of SCCmec types have been reported in Korea from type I to type VIII [14, 32, 40, 50]. However, there are no reports about the detection of other pressure VI yet. Strains reported in Korea have been reported mainly on recent emergence of new mutations in existing types. In addition, types II, III, and IV are detected among existing types.

This study and results reported in Korea reveal that the SCCmec type mainly shows multi-drug resistance. while the degree of resistance was weak for type I, the distribution of type III showed some a part focused.

This study should be carried out continuously in the future.

Thank you very much

Submission Date

04 August 2023

Date of this review

16 Aug 2023 00:06:33

Open Review

Quality of English Language

(x) I am not qualified to assess the quality of English in this paper
( ) English very difficult to understand/incomprehensible
( ) Extensive editing of English language required
( ) Moderate editing of English language required
( ) Minor editing of English language required
( ) English language fine. No issues detected

Yes

Can be improved

Must be improved

Not applicable

Does the introduction provide sufficient background and include all relevant references?

(x)

( )

( )

( )

Are all the cited references relevant to the research?

(x)

( )

( )

( )

Is the research design appropriate?

(x)

( )

( )

( )

Are the methods adequately described?

( )

(x)

( )

( )

Are the results clearly presented?

( )

(x)

( )

( )

Are the conclusions supported by the results?

(x)

( )

( )

( )

Comments and Suggestions for Authors

The article is well written and devoted to the relevant topic of the spread of antimicrobial resistance determinants in Staphylococcus aureus, a dangerous pathogen of nosocomial infections.   The authors' own data are presented not in isolation, but against the background of literature analysis, which helps to evaluate the processes of molecular epidemiology of Staphylococcus aureus holistically.  Nevertheless, there are a number of small remarks.

Line 56: erroneous hyperlink to pharmacy website instead of mlst website. Line 41: similar. In general, you should double-check all hyperlinks in the text

Response #1. Thank for your careful consideration of the manuscript.

I have changed web site.

Sequences of our results are compared to known alleles at each locus via the MLST website (https://pubmlst.org/organisms/staphylococcus-aureus),

Line 129: "by" instead of "my"

Response #2. Thank for your careful consideration of the manuscript.

I have changed the part.

“by”

Line 139: "SCCmecA" instead of "SCCmec A"

Response #3. Thank for your careful consideration of the manuscript.

I have changed the part.

SCCmecA

Line 50: invalid link

(https://www.sccmec.org/index.php/en)

Line 137: "It" instead of "I"

Response #4. Thank for your careful consideration of the manuscript.

I have changed the part.

“I”

Line 102-103: incorrectly listed primer ends (5' instead of 50 and 3' instead of 30)

Response #5. Thank for your careful consideration of the manuscript.

I have changed the part.

(5’-AGACGATCCTTCGGTGAGC-3’) and spa1517R (5’-GCTTTTGCAATGTCATTTACTG-3’).

Line 146: The abbreviation gaps looks really bad. Please,  make the font smaller in Table 1 so that words are not moved to another line

Response #6. Thank for your careful consideration of the manuscript.

I have changed the part.

Line 146: It is not clear which MLST types were in isolates Gh6, Gh13, Gh38, Gh49, Gh54, Gh65, Gh68, Gh84, Gh90 in table1 and 2.  

Response #7. Thank for your careful consideration of the manuscript.

I have changed the part.

A total of 134 S. aureus strains were obtained from clinical patients at Gachon University Gil Medical Center

Than this samples are Unknown 9 mlst new types samples.

To understand the epidemiologic picture, it would be good to clarify. 

Line 155: gil should be capitalized as in the text earlier - it is the name of the hospital

Response #8. Thank for your careful consideration of the manuscript.

* gh is 9/134 new types samples in “Gil” hospital

Line 194 and 196: MLST should be capitalized.

Response #9. Thank for your careful consideration of the manuscript.

I have changed the part.

“MLST”

Page 6: picture and caption on different pages (not very readable). Maybe figures 1 and 2 would be better combined

Response #10. Thank for your careful consideration of the manuscript.

This figure will be expected to be edited.

Line 217: the word "Types" should be written with a lowercase "t"

Response #11. Thank for your careful consideration of the manuscript.

I have changed the part.

After that reports decreased. Around 2005, there were many reports of III, IIIA, and IV types [34-36].

Line 228: "A greater diversity" should be preceded by a period, not a comma. Or start with a small letter instead of a capital letter.

Response #12. Thank for your careful consideration of the manuscript.

I have changed the part.

a greater diversity

Line 255: "while the degree" with a capital "W".

Response #13. Thank for your careful consideration of the manuscript.

I have changed the part.

While the degree

Line 260: "SCCmec type. [14,15,40]." One dot is unnecessary

Response #14. Thank for your careful consideration of the manuscript.

I have changed the part.

SCCmecII types and ST72 type has many SCCmecIV types [38, 9].

Line 261: "ST 5" should be written without a space.

Response #15. Thank for your careful consideration of the manuscript.

I have changed the part.

that ST5, ST72

Line 217: "thatreports" insert a space.

Response #16. Thank for your careful consideration of the manuscript.

I have changed the part.

After that reports decreased. Around 2005, there were many reports of III, IIIA, and IV types [34-36].

Line 201: should be written MLST, not "mslt"

Response #17. Thank for your careful consideration of the manuscript.

I have changed the part.

“MLST”

It would be interesting to see the phylogenetic distribution patterns of SCCmec across all Korean MLST types of Staphylococcus aureus. 

What software and with what parameters were used for phylogenetic figures 1 and 2 ?

Submission Date

04 August 2023

Date of this review

16 Aug 2023 12:57:26

Reviewer 2 Report

The article is well written and devoted to the relevant topic of the spread of antimicrobial resistance determinants in Staphylococcus aureus, a dangerous pathogen of nosocomial infections.   The authors' own data are presented not in isolation, but against the background of literature analysis, which helps to evaluate the processes of molecular epidemiology of Staphylococcus aureus holistically.  Nevertheless, there are a number of small remarks.

Line 56: erroneous hyperlink to pharmacy website instead of mlst website. Line 41: similar. In general, you should double-check all hyperlinks in the text

Line 129: "by" instead of "my"

Line 139: "SCCmecA" instead of "SCCmec A"

Line 50: invalid link

Line 137: "It" instead of "I"

Line 102-103: incorrectly listed primer ends (5' instead of 50 and 3' instead of 30)

Line 146: The abbreviation gaps looks really bad. Please,  make the font smaller in Table 1 so that words are not moved to another line

Line 146: It is not clear which MLST types were in isolates Gh6, Gh13, Gh38, Gh49, Gh54, Gh65, Gh68, Gh84, Gh90 in table1 and 2.  

To understand the epidemiologic picture, it would be good to clarify. 

Line 155: gil should be capitalized as in the text earlier - it is the name of the hospital

Line 194 and 196: MLST should be capitalized.

Page 6: picture and caption on different pages (not very readable). Maybe figures 1 and 2 would be better combined

Line 217: the word "Types" should be written with a lowercase "t"

Line 228: "A greater diversity" should be preceded by a period, not a comma. Or start with a small letter instead of a capital letter.

Line 255: "while the degree" with a capital "W".

Line 260: "SCCmec type. [14,15,40]." One dot is unnecessary

Line 261: "ST 5" should be written without a space.

Line 217: "thatreports" insert a space.

Line 201: should be written MLST, not "mslt"

It would be interesting to see the phylogenetic distribution patterns of SCCmec across all Korean MLST types of Staphylococcus aureus. 

What software and with what parameters were used for phylogenetic figures 1 and 2 ?

Author Response

(The authors gave the same response as above.)
